# Experimental Investigations of the Thermal Safety of Methyl Ethyl Ketone Oxime Hydrochloride Based on the Flask Method, Thermal Analysis, and GC-MS

**Dehong Zhou, Shiyu Peng \* , Bin Xie, Lunping Wang and Haochen Li**

School of Resource & Safety Engineering, Wuhan Institute of Technology, Wuhan 430074, China; dhzhou@wit.edu.cn (D.Z.)
* Correspondence: 22117010015@stu.wit.edu.cn; Tel.: +86-139-390-13106

**Abstract:** Chemical safety accidents caused by the thermal runaway of materials occur frequently around the world, seriously hindering the sustainable development of the chemical industry. Therefore, studies related to the thermal safety of materials are very important for chemical production. In order to ensure the safety of methyl tris (methyl ethyl ketone oxime) silane (MOS), the thermal safety of its accident-prone by-product, methyl ethyl ketone oxime hydrochloride (MEKOH), was analyzed in the study. Temperature changes of MEKOH dissolved in 5%, 10% and 15% deionized water were measured with the flask method. Thermogravimetric (TG) analysis and differential scanning calorimetry (DSC) were applied to comprehensively analyze the thermal stability of MEKOH in different reaction states. The thermal decomposition products of MEKOH were detected with gas chromatography-mass spectrometry (GC-MS). The results show that the temperature of MEKOH dissolved in deionized water at room temperature (28 °C) increases by about 5 °C, and finally stabilizes at 33 °C. MEKOH has good thermal safety during this process. When the temperature rises to 50 °C, MEKOH starts to decompose violently, and no longer exhibits significant weight loss at 145 °C. From 50 °C to 100 °C, MEKOH releases heat, ranging from 29.65 to 45.86 J/g, during thermal decomposition, generating a large amount of flammable substances. The thermal decomposition products were detected, including pyrrolidine, heptane, MEKO, and other substances, but no MEKOH was detected. Overall, the study provides a theoretical basis for preventing the thermal runaway of MEKOH.

**Keywords:** thermal safety analysis; thermal stability; flask method; thermogravimetric analysis; differential scanning calorimetry; gas chromatography analysis; mass spectrometry

## 1. Introduction

The safety of chemical production determines the sustainability of the development of the chemical industry. In recent years, chemical safety accidents have occurred frequently around the world, seriously hindering the stable and healthy development of the chemical industry [1]. As China is the world's largest chemical country, chemical safety has become an important issue in China's industrial development [2]. According to relevant statistics, a large number of chemical safety accidents are currently caused by the thermal runaway of materials [3]. Such accidents have a certain abruptness, and always have serious consequences including casualties, property damage, environmental destruction, and production stagnation [4,5]. Therefore, securing the thermal safety of materials and effectively improving the whole safety level of chemical enterprises have become urgent problems to be solved to ensure the long-term safe development of the industry [6].

Take Hubei Province as an example. In August 2020, a chemical safety accident occurred during MOS production. A flash explosion occurred in the static tank, resulting in a total of six deaths and four injuries. The accident scene is shown in Figure 1. According to the relevant provisions of the *Work Safety Law of the People's Republic of China* and *Chinese Regulations on the Reporting, Investigation, and Disposition of Work Safety Accidents*, an accident

that causes "3 to 10 deaths, 10 to 50 serious injuries, or direct economic losses of 10 million to 50 million yuan" is defined as a "major accident". This accident was the most serious accident in Hubei Province since 1998.

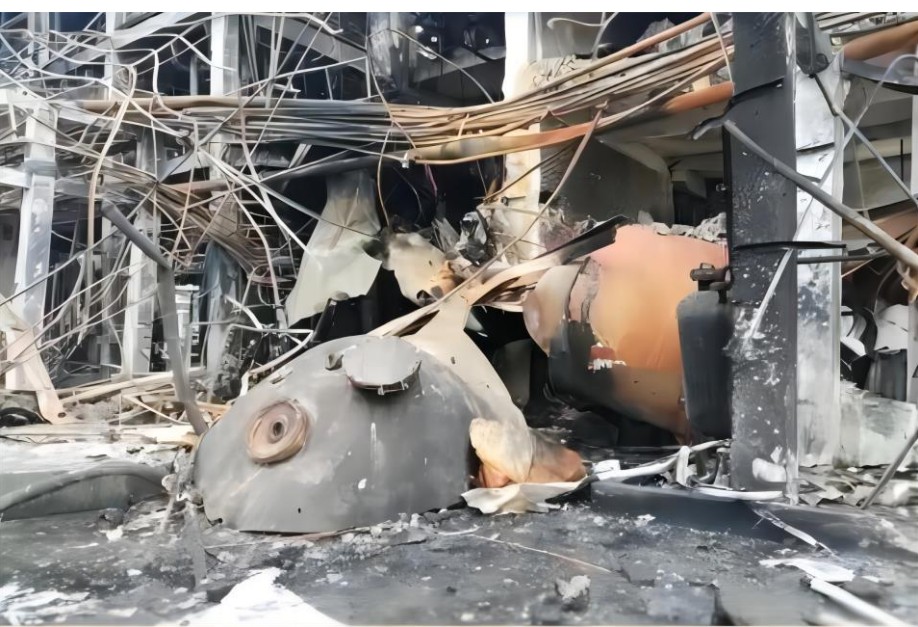

**Figure 1.** Flash explosion accident scene.

Based on the field investigation and analysis, the accident was caused by the incomplete cleaning of the stratification tower and the decomposition of a large amount of MEKOH in the static tank. According to the relevant literature [7], MEKOH is prone to decomposition at 50 to 70 °C and may even lead to fire and explosion accidents. In the accident, the temperature of the resting tank where the accident occurred was found to exceed the acceptable range for MEKOH, leading to violent decomposition of the substance and a flash explosion.

As MOS has been introduced in several industries, the synthesis process optimization and work safety analysis of MOS and related products have gradually attracted the attention of scholars. Nelson et al. [8] applied the ASTM F739-96 standard [9] testing method to test the penetration resistance of gloves and protective clothing barriers of some marketable alternative silanes and siloxanes, including MOS. Li et al. [10] summarized the common uses and main synthetic methods of MEKO and evaluated the feasibility of the commonly used methods, pointing out research directions on how to reduce production costs and optimize the synthesis process. Zhao et al. [11] synthesized a new series of autocatalytic cross-linking agents for RTV silicone rubber and α-amine ketoxime silanes. Compared with MOS, RTV silicone rubber was better in terms of mechanical properties and viscosity. Chen et al. [12] prepared multifunctional coatings with both superhydrophobicity and high conductivities by mixing MOS and other materials, which can be widely used in many fields.

Through a review of the previous literature, the following conclusions are obtained. Although a few studies on MOS and related products have been proposed at home and abroad, the relevant research is still in the initial stage. A certain gap still exists between experimental research and actual production, and industrial production cannot be fully guided by scientific theories. Moreover, current research is mostly focused on optimizing the synthesis process of MOS and improving the production and utilization efficiency of the materials. Studies on the safety of MOS and its by-products are still lacking.

In the production process of MOS, MEKOH is synthesized in large quantities. MEKOH is highly dangerous and can easily cause fire and explosion accidents once the temperature is too high. Through an analysis of the actual production process, MOS work safety

accidents are probably caused by the exothermic decomposition of the by-product MEKOH. To improve the safety of industrial production and avoid the recurrence of such accidents, the research on the thermal safety of MEKOH is of great practical significance.

With reference to a large number of domestic and international experiments and studies on the thermal safety of chemical materials, current thermal behavior experiments with high applicability and accuracy were summarized to include thermal analysis experiments and GC-MS experiments [13–15]. The commonly used thermal analysis techniques include thermogravimetric (TG) analysis [16], differential scanning calorimetry (DSC) [17], differential thermal analysis (DTA), dynamic mechanical analysis (DMA) [18], and thermal mechanical analysis (TMA) [19]. Among them, TG analysis was one of the first thermal analysis techniques to be discovered and applied, and is now used in a wide range of fields such as materials science [20–22], agriculture and forestry [23,24], and geology [25]. The method was first applied in the 1880s. Higgins [26] first used a balance to weigh the weight changes that occurred when the sample was heated while analyzing lime binder and quicklime. Afterward, TG analysis began to be gradually applied as an analytical tool. In 1915, Japanese scholars used the thermal equilibrium method to analyze the weight loss process of inorganic compound thermal decomposition reactions. In the 1950s, American scholars conducted a series of studies on TG analysis, which then began to be applied in many fields.

Gas chromatography (GC) is a technique for analyzing the composition of samples by chromatography and detection, and is a type of separation science [27]. GC has a wide range of applications in many areas including food safety [28,29], chemical analysis [30], and environmental monitoring [31,32]. Although the application of GS-MS in China is in the initial stage compared to that of developed countries, with the continuous advancement of relevant research and topics, it has made certain breakthroughs in many fields [33]. Especially in the research of instruments, the analytical accuracy has reached a world-class level. Therefore, GC-MS has a very broad development prospect in China.

Previous literature statistics revealed that there are currently no relevant studies on the thermal safety of MEKOH. Therefore, the flask method, a thermal analysis experiment, and the GC-MS method were applied to comprehensively analyze the thermal safety of MEKOH in this paper. The main objectives were to analyze the thermal stability and thermal behavior of MEKOH through the above experiments and to determine the temperature change of MEKOH dissolution, as well as the temperature threshold and pyrolysis products of the violent thermal decomposition. The thermal safety of MEKOH was comprehensively analyzed by combining the above experimental results. From the application point of view, this study was conducted to provide a theoretical basis for the prevention of MEKOH thermal runaway accidents, to improve the safety of chemical production in organosilicon enterprises to a certain extent, and to promote the long-term stable development of enterprises.

## 2. Material Analysis

### 2.1. Characterization

MOS is a kind of compound with the CAS Registry Number 22984-54-9. The substance has a boiling point of 310 °C, a melting point of −22 °C, a flash point of 61 to 93 °C, a refractive index of 1.4578, and a pH range of 6 to 9.5. At room temperature, MOS is a colorless or slightly yellow transparent liquid with the odor of MEKO. At 25 °C, the density of MOS is 0.975 ± 0.005. When the temperature drops to 20 °C, the relative density of MOS increases to 0.982 [34]. When exposed to heat, MOS will decompose and release nitrogen oxide ($NO_x$), silica, carbon dioxide, carbon monoxide, butanone, and possibly MEKO.

In recent years, with the increasing application of silicone in various industries, the market for silicones has been expanding, with the consumption and production rising year on year (see Figure 2). As an important component affecting the quality of silicone sealants [35], the demand for MOS continues to increase. In industrial applications, MOS has good market application prospects. MOS is frequently used as a neutral cross-linking

agent for room-temperature vulcanized (RTV) silicone rubbers and silicone glass adhesives, and is also widely used in many industries, including construction, electronics, pharmaceuticals, and automotive manufacturing [36,37].

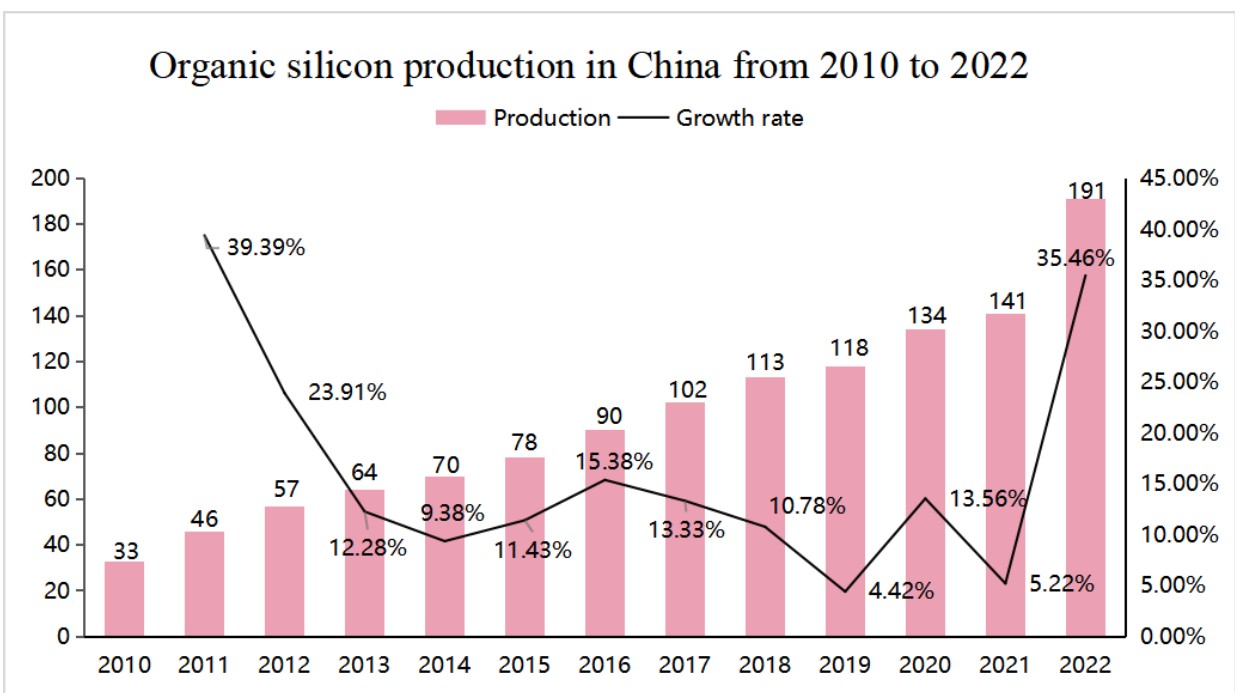

**Figure 2.** China silicone production from 2010 to 2022.

### 2.2. Preparation of MOS

MOS was initially synthesized by the reaction of chlorosilanes and oxime compounds, with certain amounts of organic bases such as triethylamine as a deacidification reagent [7]. Although the production process has the advantages of low production costs and high productivity, the distillation and separation of products during the production process easily lead to explosions. Later, Muiller et al. [38] proposed a new method for preparing silicon compounds without using acid-binding agents, in which MEKO and compounds containing Si–N bonds could be synthesized into MOS. However, the process could not be practically applied in industrial production due to its complexity and the fact that the raw materials were not readily available.

As the chemical industry continues to develop, the MOS production process is being optimized. Common MOS production processes currently include titration processes, transesterification [39], carboxyl exchange, and sodium metal synthesis. Due to the high production costs, the difficulty in obtaining raw materials, the complexity of the production process, and the low safety coefficient of the latter three processes, the methods are rarely used in industrial production [34].

Due to space limitations, this paper only presents a short description of the most common synthetic processes currently used in industrial production. The most widely used method in industrial production now is chlorosilane titration, which is a direct titration method. The method can be subdivided into two specific synthesis processes, namely intermittent titration and continuous titration. Intermittent titration is the most commonly used synthesis process.

The intermittent titration method is used to produce MOS by reacting chlorsilane and MEKO [7,40–42]. The method includes reaction, separation, neutralization, standing, extraction, and distillation [43]. In actual production, chlorsilane is added dropwise to an excess of MEKO to undergo an oximation reaction (see Equation (1)). The oximation reaction is an exothermic reaction in which large amounts of MEKOH are produced. After

the reaction, the reaction product is left to split into two layers; the upper layer consists of crude MOS, MEKO, and most of the solvent, while the lower precipitate is MEKOH and a small amount of crude MOS. Then, ammonia (NH$_3$) is introduced to neutralize the upper crude MOS. The neutralized solution is left to settle, the lower solids are removed by filtration, and the upper crude layer is used in the distillation process.

$$CH_3SiCl_3 + 6HNO = CCH_3C_2H_5 \rightarrow CH_3Si(ON = CCH_3C_2H_5)_3 + 3HCl \cdot HNO = CCH_3C_2H_5 \tag{1}$$

A small amount of MEKOH also remains in the upper reaction product of the oximation reaction. MEKOH can undergo side reactions while neutralizing MOS in NH$_3$, generating solid ammonium chloride (NH$_4$Cl), as shown in Equation (2).

$$HCl \cdot HNO = CCH_3C_2H_5 + NH_3 \rightarrow HON = CCH_3C_2H_5 + NH_4Cl \tag{2}$$

The lower reaction product is extracted with the solvent and the extraction solvent containing the final MOS product is collected for recycling. The lower layer of MEKOH is neutralized with aqueous ammonia in an oxime salt neutralization circulation system, the reaction mixture is separated by settling, and the upper layer of aqueous MEKO is distilled to obtain pure MEKO. Finally, the upper crude product obtained by filtration in the second step is evaporated, the evaporated solvent is recycled, and the final MOS product is obtained at the bottom of the evaporator kettle.

The specific production process is summarized in Figure 3.

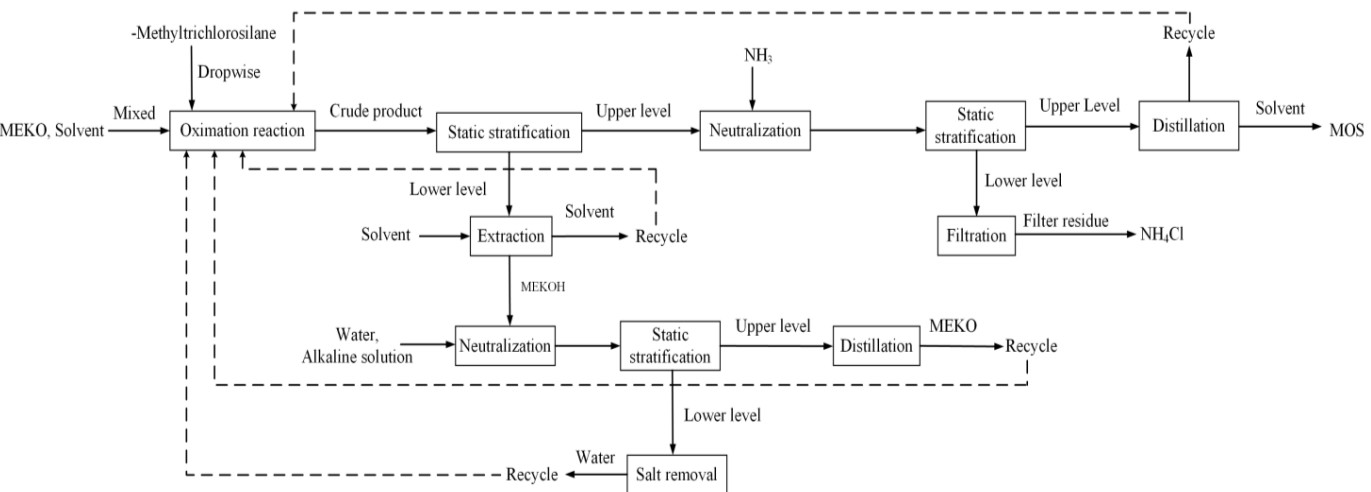

**Figure 3.** MOS manufacturing process.

### 3. Experimental Process

#### 3.1. Specific Experimental Materials and Equipment

Approximately 200 mL of MEKOH was chosen as the sample for the experiment, and was numbered Sample 01, Sample 02, Sample 03, and Sample 04. MEKOH for the experiment was supplied by the relevant company. Experimental supplies include 10 mL and 100 mL graduated mill-mouth flasks, a mechanical stirrer, a thermostat with an accuracy of $\pm 0.5\,°C$, a centrifuge, a temperature controller, a sensor, and a cooling unit. All laboratory supplies were provided by the Chemical Analysis and Testing Centre of Wuhan Institute of Technology, and all meet the requirements of the relevant specifications.

The detailed specifications of the experimental apparatus are shown in Table 1.

**Table 1.** Experimental instrument parameters.

| Experiment | Relevant Bas | Equipment | Type | Grade | Number |
|---|---|---|---|---|---|
| Flask method | GB/T 27841-2011 Chemical products for industrial use—Determination of hydrosolubility of solids and liquids with high solubility—Flask method [44] | Graduated flask with ground mouth | 100 mL | Compliant with ISO 4788 [45] | A01-A10 |
| Thermal analysis | GB/T 13464-2008 Thermal analysis test methods for thermal stability of materials [46] | NETZSCH synchronous thermal analyzer | STA449F3 | Qualified | 1379-M |
| GC-MS | GB/T 9722-2006 Chemical reagent—General rules for gas chromatography [47] | Agilent gas chromatography-mass spectrometry | GC 7890A MSD 5975C | Qualified | CN10481054 |

*3.2. Experimental Procedure*

3.2.1. Flask Method Experiment

The flask method is a commonly used solubility measurement method recommended by the Organization for Economic Cooperation and Development (OECD) [48,49]. The principle is that the MEKOH sample is dissolved in water under experimental conditions slightly above the measurement temperature until saturation is reached. Next, the solution is cooled to capture and analyze the temperature change of the solution during dissolution [50]. The method is widely used because of its simple operation, high reliability, and ease of implementation in the laboratory.

According to *Chemical products for industrial use—Determination of hydrosolubility of solids and liquids with high solubility—Flask method* [44], 2.5 g, 5 g, and 7.5 g of deionized water were dissolved in approximately 50 g of Sample 01. That is, the thermal effect of 5%, 10%, and 15% dissolved deionized water samples of Sample 01 was tested. Firstly, roughly 50 g of sample was added to a 100 mL flask and 2.5 g of deionized water was added gradually at room temperature. The temperature of the solution was measured and recorded before dissolving. Then, the flask was corked and placed in a thermostat and the mixture was stirred with a magnetic stirrer until it was completely dissolved. After complete dissolution, the flask was removed and the temperature was measured again. Following the same method, 5 g and 7.5 g of deionized water were tested, followed by Sample 02, Sample 03, and Sample 04, respectively.

3.2.2. Thermal Analysis Experiment

Thermal analysis is a method of analyzing the thermal stability of a substance, and is used to characterize the relationship between the properties of substances and the temperature [51,52]. The thermal analysis methods applied in this paper include TG analysis and DSC. TG analysis is mainly used to study the thermal changes accompanied by a mass increase or decrease after decomposition, combination, dehydration, evaporation, and other processes [53,54]. Based on this method, qualitative analysis, component analysis, thermal parameter determination, and kinetic parameter determination can be performed on substances [55].

Since TG analysis can only measure data related to mass changes, it cannot study physical phenomena without a mass change (such as melting and crystallization), so comprehensive analysis with other thermal analysis techniques is necessary. Another thermal analysis technique applied in this paper is DSC. It is a method of measuring the variation process of the heat flux power difference between a sample and a reference material as a function of temperature (or time). The method can be applied to analyze the specific heat capacity, reaction heat, transition heat, and other thermodynamic and kinetic parameters. With the advantages of a wide temperature range (−175~725 °C), high

resolution, and low sample consumption, DSC is widely used for thermal stability analyses of some processes, including alloy phase transformations [56], fossil fuel combustion [57], and other processes.

Based on *Thermal analysis test methods for thermal stability of materials* [46], the NETZSCH synchronous thermal analyzer (model STA449F3) was used for TG-DSC analysis of Sample 01 in this experiment. The combined application of TG analysis and DSC can obtain the mass change and thermal effect data of the sample through a single measurement [58]. The energy change of the sample during the thermal reaction process is reflected in DSC. The mass change of the sample during the thermal reaction is characterized in TG analysis, which can be used to quantitatively calculate the composition of the substance [59]. To ensure the scientific validity of the results, the experiments were repeated three times.

The temperature range for the experiment is 30 °C to 200 °C and the temperature rise rate was 10 °C/min. To exclude the interference of oxygen with the experiment, nitrogen was selected as a protective atmosphere, with a flow rate of 60 mL/min. The analysis steps were as follows.

Before the experiment, the apparatus was calibrated for both temperature and heat flow with an accuracy of $\pm 0.5$ °C. Then, 10 mL of Sample 01 was taken and used in the experiment. Both the obtained specimen and the reference material were placed in their respective sample containers with good contact heat with the containers, and 20% of the sample weight of inert material was added. Thereafter, the sample containers were placed together in the heating unit of the instrument and brought into close contact with the sensing element. $N_2$ protective gas was switched on and the gas flow rate was controlled at around 60 mL/min. After ensuring that the normal working conditions of the instrument are at room temperature and atmospheric pressure, the temperature controller was activated to control the rate of temperature increase from 2 °C/min to 20 °C/min, with a heating range of 30 to 200 °C. Then, the relationship curve between power difference and temperature T was recorded, which is the DSC curve.

Samples were tested three times following the above steps, and the average of three measurements was taken as the results. The difference between the three measurements should be guaranteed to be within the accuracy range. If it is not within the accuracy range, the above procedures need to be repeated until the test is accurate.

### 3.2.3. Gas Chromatography-Mass Spectrometry Experiment

In the study, a combination of gas chromatography (GC) and mass spectrometry (MS) techniques was applied to qualitatively analyze the MEKOH samples. Compared to a single method, GC-MS analysis can effectively combine substance separation and qualitative analysis, improving the efficiency of experiments and the accuracy of conclusions [60,61]. According to *Chemical reagent—General rules for gas chromatography* [47], Sample 02, Sample 03, and Sample 04 were analyzed with an Agilent Gas Chromatography-Mass Spectrometer (GC 7890A, MSD 5975C). The specific experimental principles are as follows.

The sample and its measured components were vaporized and introduced into the column simultaneously with the carrier gas. Depending on the difference in the physical and chemical properties of the components to be measured between the gas and solid or gas and liquid phases, the substances were separated in the column due to the difference in the migration rate of the components [62]. After separation, neutral molecules were eluted through the transmission line into the mass spectrometer. In the mass spectrometer, the separated molecules were ionized by electrons and cleaved into molecular ions and free radical cations [63]. Due to differences in molecular formulae, molecular structures, and broken bonds, the molecular ions may be rearranged or fragments may be lost. Ions with different masses were separated depending on the mass-to-charge ratio and recorded by the data processing system to produce a chromatogram and a mass spectrum.

Considering that MEKOH is insoluble in dichloromethane, diethyl ether, methyl ether, toluene, and ethanol, the experimental protocol was changed to dissolve the sample in a clear solution ($C_{14}H_{27}N_3O_3Si$) by high-speed stirring (500 rpm for 12 h). The analytical

conditions for this experiment were an injection volume of 1 μL, an injector temperature of 180 °C, and a carrier gas flow rate of 20 mL/min. The temperature rise program was 30–250–315 °C, with a temperature rise rate 10 °C/min. The range for detecting molecular weights is 5 to 450. The test was repeated three times for each sample to ensure the scientific validity of the results.

## 4. Results

### 4.1. Water Solubility Analysis of MEKOH

The MEKOH temperature rise variation curves obtained from flask experiments on four sets of samples are shown in Figure 4. The temperature variation of the exothermic dissolution of Sample 01 generally stabilized by the eighth minute of the experiment. During dissolution, the stable temperature and starting steady time differed depending on the quantity of the deionized water. For the experiment with 5% deionized water, the temperature was 31.5 °C at 2.5 min, and the rate of increase in temperature decreased significantly between 2.5 min and 8 min. By the eighth minute, the temperature was 32.5 °C and no further significant changes occurred. For 10% deionized water, the rate of temperature increase was found to slow down significantly after 4 min and the solution temperature was stable at 32 °C by 8 min into the experiment. The 15% deionized water dissolution experiment was stable at 31 °C at the eighth min of reaction.

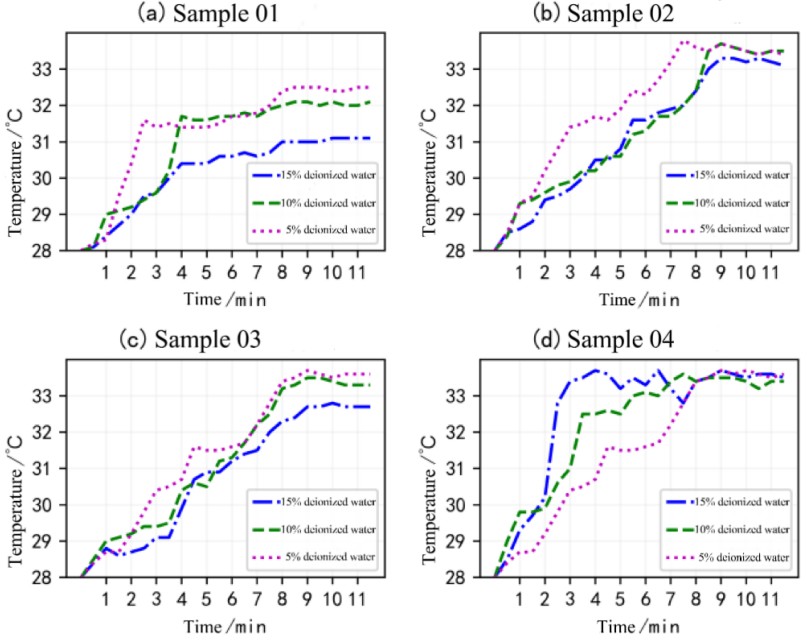

**Figure 4.** MEKOH dissolution temperature rise curve.

Figure 4b was obtained from the experiment of Sample 02, where the exothermic temperature of the sample dissolved in 5% deionized water, 10% deionized water, and 15% deionized water stabilized at 33.5 °C, 33.5 °C, and 33.0 °C, respectively, at the ninth minute of the experiment. In particular, the 5% deionized water sample started to stabilize after 7.5 min of the experiment. According to Figure 4c, the exothermic temperature of Sample 03 dissolved in 5% deionized water, 10% deionized water, and 15% deionized water stabilized at the 10th min, with stabilization temperatures of 33.5 °C, 33.2 °C, and 32.7 °C, respectively.

For the fourth group of samples, although the percentage of deionized solutions was different, the temperatures stabilized at around 33.5 °C. The 5% deionized water continued to be exothermic until the ninth min, and the temperature began to stabilize at 33.5 °C. The 10% deionized water sample reached 33.5 °C at the 7.5th min, after which the temperature data only showed small fluctuations with no significant changes. The temperature first

reached 33.5 °C when 5% deionized water was used for 4 min. However, the reaction was unstable between 4 min and 9 min, with the temperature dropping to 33.8 °C at 7.5 min. At the ninth min, the temperature finally stabilized at 33.5 °C.

The specific experimental data are shown in Table 2.

**Table 2.** Data from flask experiment.

| No. | Percentage of Deionized Water/% | Initial Temperature/°C | Stable Temperature/°C | Start Stable Time/min | Temperature Contrast/°C |
|---|---|---|---|---|---|
| Sample 01 | 5 | 28.0 | 32.5 | 2.5 | 4.5 |
| | 10 | 28.0 | 32.0 | 4 | 4.0 |
| | 15 | 28.0 | 31.0 | 8 | 3.0 |
| Sample 02 | 5 | 28.0 | 33.5 | 7.5 | 5.5 |
| | 10 | 28.0 | 33.5 | 9 | 5.5 |
| | 15 | 28.0 | 33.0 | 9 | 5.0 |
| Sample 03 | 5 | 28.0 | 33.5 | 10 | 5.5 |
| | 10 | 28.0 | 33.2 | 10 | 5.2 |
| | 15 | 28.0 | 32.7 | 10 | 4.7 |
| Sample 04 | 5 | 28.0 | 33.5 | 4 | 5.5 |
| | 10 | 28.0 | 33.5 | 7.5 | 5.5 |
| | 15 | 28.0 | 33.5 | 9 | 5.5 |

As can be seen from Table 2, the temperature change after MEKOH is dissolved in deionized water is about 5 °C, indicating that the heat release from MEKOH dissolution will not cause chemical accidents.

### 4.2. TG-DSC Analysis of MEKOH

TG-DTG and DSC curves obtained in the thermal analysis experiment with Sample 01 are shown in Figures 5–8. The thermal analysis curves for the initial, intermediate, and complete states of the MEKOH decomposition process are depicted in the figures. The weight loss ratios of the decomposition and the char yield of samples are shown in the TG curves. The peaks and troughs of the DTG curve characterize the temperature at which the rate of weight loss was fastest and slowest, which generally correspond to the peaks of energy change in the DSC curve. The heat change during the reaction could be visualized in the DSC curve, and the absorption or exothermic effect of a reaction was represented by a 'peak'. Combining the DSC and DTG curves, the temperature bands corresponding to the different thermal changes were further determined. Combined with the TG curve, the change in mass of the samples over the temperature range could be determined.

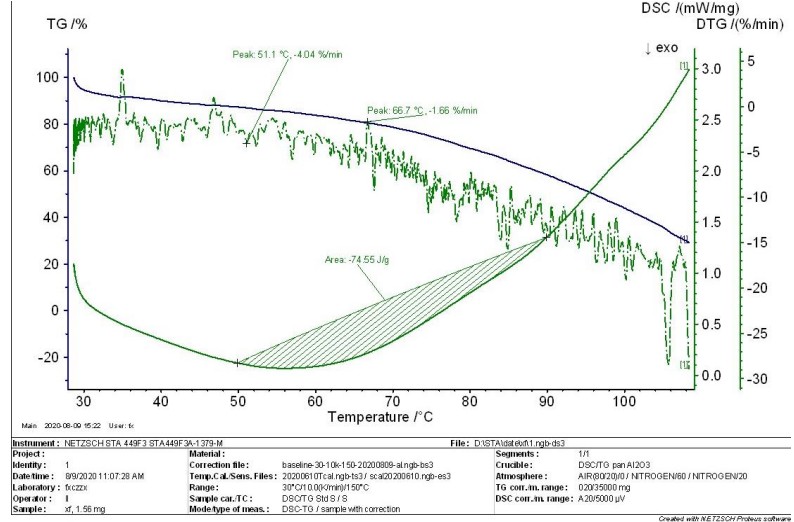

**Figure 5.** TG−DTG−DSC curve in the initial state.

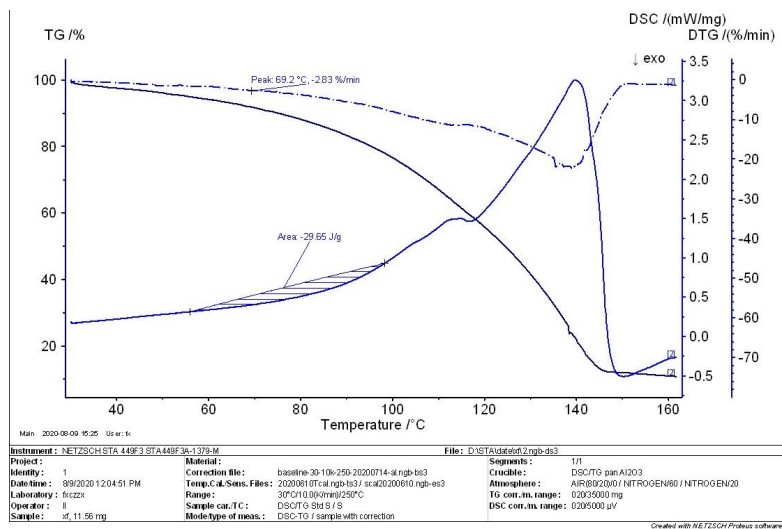

**Figure 6.** TG−DTG−DSC curve I in the intermediate state.

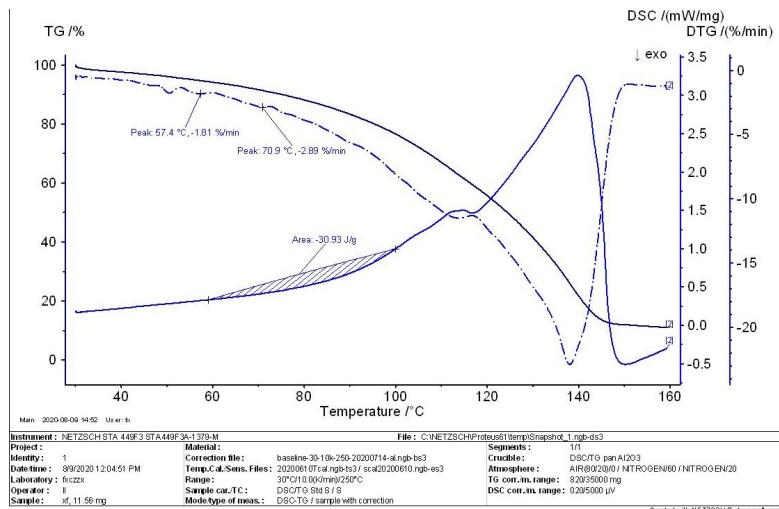

**Figure 7.** TG−DTG−DSC curve II in the intermediate state.

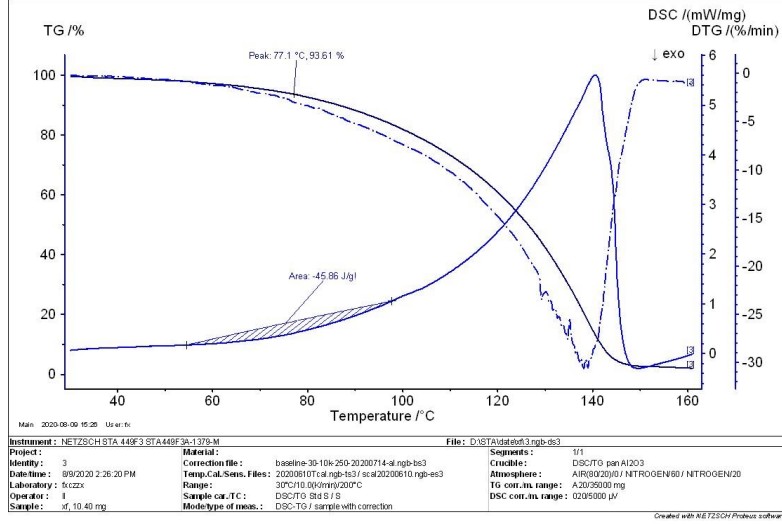

**Figure 8.** TG-DTG-DSC curve in the complete state.

The curves in Figure 5 show that the char yield of the MEKOH sample is 30%, and the heat release of the sample from 50 °C to 90 °C is 74.55 J/g. In addition, maximum peaks appear at 51.1 °C and 66.7 °C, respectively, and the enthalpy increases in this range, which means that an exothermic reaction occurs.

The thermal analysis curves of the sample in the intermediate state are shown in Figures 6 and 7. The initial maximum peak for the decomposition of the MEKOH sample in Figure 6 corresponds to a temperature point of 69.2 °C. The residual of the sample at this temperature point is 92.5%. Additionally, the first derivative was calculated to give a weight loss rate of 2.83%/min. The heat release of the sample from 58 °C to 98 °C was calculated to be 29.65 J/g. In Figure 7, the two largest peaks were found in the decomposition of MEKOH, corresponding to temperature points of 57.4 °C and 70.9 °C and weight loss rates of 1.81%/min and 2.89%/min. The heat release of the sample in such a state was calculated to be 30.93 J/g.

When the decomposition reaction of MEKOH reached completion, the sample was analyzed for thermal stability, and the curve in Figure 8 was obtained. The char yield shown in the TG curve is approximately 3%, indicating that MEKOH is almost completely decomposed. The peak of MEKOH in this state is 77.1 °C, and the rate of weight loss is 93.61%. The enthalpy of reaction from 55 °C to 98 °C is 45.86 J/g.

A comprehensive analysis of the thermal stability of the decomposition of MEKOH shows that the reaction first occurred at maximum peak at temperature points of 51.1 °C and 57.4 °C. This result illustrates that the sample began to lose weight between 51 °C and 57 °C, meaning that MEKOH started to decompose at 51 °C to 57 °C. Analyzing the amount of heat released in the intermediate and complete states, the reaction enthalpy of MEKOH between 55 °C and 100 °C was found to range between 29.65 and 45.86 J/g.

To fully reflect the weight loss during the decomposition of MEKOH, TG curves for the three states are summarized in Figure 9. The trajectories of the two TG curves [2] for the intermediate state are almost coincident, and the char yields are both close to 10%; thus, the TG curves of the intermediate state can be assumed to be almost identical. To reduce the complexity of the figure, only one intermediate state curve is shown. Analysis of the curves concludes that MEKOH decomposition results in a char yield of 30% in the initial state, approximately 10% in the intermediate state, and 3% in the complete state.

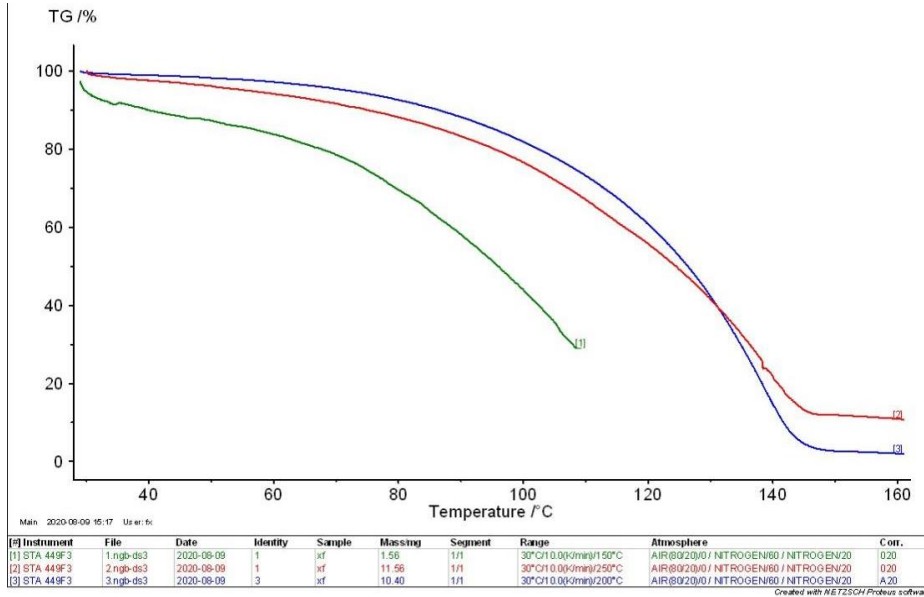

**Figure 9.** TG curves for different states.

### 4.3. Analysis of MEKOH Decomposition Substances

Based on the differences in physical properties and structures of the different components of the mixture, the types and masses of substances produced by the decomposition of Sample 02, Sample 03, and Sample 04 were investigated with CG-MS, and gas chromatograms and mass spectrograms were obtained, respectively. The experiment was conducted in nine groups and the results are shown below.

Figure 10 shows the gas chromatography results for the first set of experiments. In the gas chromatogram, the abscissa is the retention time and the ordinate is the abundance. The retention values and the chromatographic peak areas or corresponding peak height values for each component are the basis for characterization and quantification, respectively [64–66]. A total of 66 peaks were measured in the first set of experiments, with four peaks having an area percentage greater than 10%. Detailed information of the generated substances is shown in Table 3.

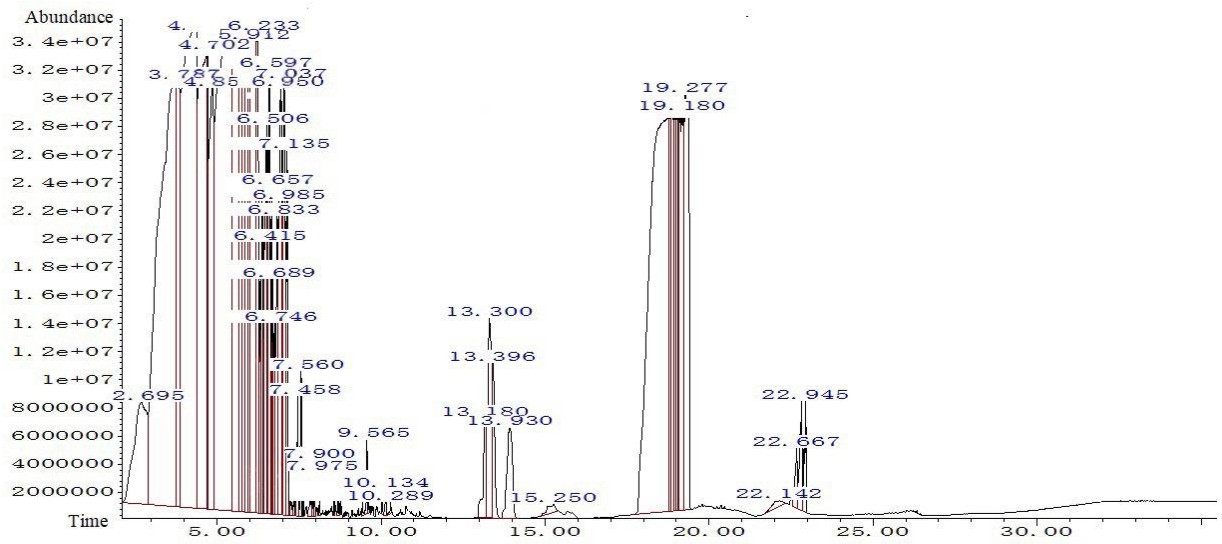

**Figure 10.** Gas chromatogram of experimental group 1.

**Table 3.** Material information with a high abundance in experimental group 1.

| Peak | Retention Time | Area | Material | Formula | CAS | Mass |
|---|---|---|---|---|---|---|
| 2 | 3.694 | 11.30 | Pyrrolidine | $C_4H_9N$ | 123-75-1 | 46 |
| | | | Hexane, 3,3,4-trimethyl | $C_9H_{20}$ | 16747-31-2 | 43 |
| | | | Heptane | C7H16 | 142-82-5 | 43 |
| 4 | 4.362 | 10.05 | Cycloheptane | $C_7H_{14}$ (isomer) | 291-64-5 | 55 |
| | | | 1-Heptene | $C_7H_{14}$ (isomer) | 592-76-7 | 50 |
| | | | Methylcyclohexane | $C_7H_{14}$ | 108-87-2 | 46 |
| 8 | 5.378 | 10.28 | Octane, 4-methyl- | $C_9H_{20}$ | 2216-34-4 | 46 |
| | | | Hexane, 2,3,4-trimethyl- | $C_9H_{20}$ | 921-47-1 | 43 |
| | | | Heptane, 2-methyl- | $C_8H_{18}$ | 592-27-8 | 38 |
| 55 | 18.746 | 11.25 | Azetidine, n-propyl- | $C_3H_7N$(isomer) | \ | 16 |
| | | | 1-Methoxy-2,3-cis-dimethylaziridine(sin) | $C_5H_{11}NO$ | 61593-25-7 | 12 |
| | | | Phosphine oxide, methyldiphenyl- | $C_{13}H_{13}OP$ | 2129-89-7 | 12 |

Figure 11 shows the mass spectrogram of HON = $CCH_3C_2H_5$ (MEKO) in experimental group 1, with a relative molecular mass of 87. The abscissa in the mass spectrogram is the mass-to-charge ratio ($m/z$). The ordinate is the abundance, which means the intensity of the ions measured in the MS analysis. According to the figure, the peak with the highest

abundance is the base peak [67], the abundance of which is $1.6 \times 10^6$. Since the mass-to-charge ratio of the base peak is the same as the relative molecular mass of MEKO, the peak is also the molecular ion peak. Unusually, the molecular ion peaks in this experiment are not the peaks with the largest mass-to-charge ratio. Such a situation is due to the combined analysis of gas chromatography and mass spectrometry and the mixing of other substances separated by GC during the MS analysis, resulting in the presence of peaks from 97.1 to 207.1.

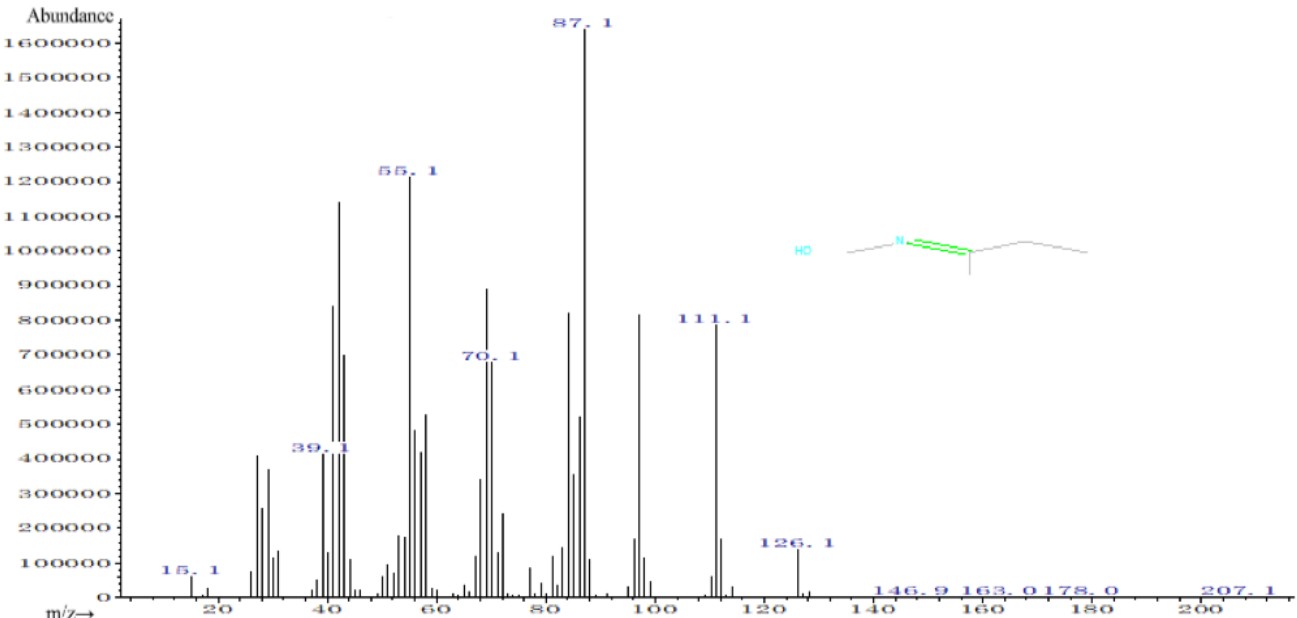

**Figure 11.** Mass spectra of experimental group 1.

Therefore, the mass-to-charge ratio of the peak on the left side of the base peak is less than the relative molecular mass, corresponding to the ion formed by electron impact. The peak to the right is formed by mixing in impurities. Only peaks to the left of the base peak were analyzed in this study. In this paper, peaks with mass-to-charge ratios of 70.1 and 55.1 are taken as examples. According to analysis, the peak with a mass-to-charge ratio of 70.1 is 17 less than the relative molecular mass of the substance, suggesting that an OH-may have been lost. The peak with a mass-to-charge ratio of 55.1 is 32 smaller than 87.1, possibly indicating breaking of an OH- and a methyl ($CH_3$).

According to the previous elaboration, the decomposition of MEKOH produces MEKO. Analysis of the substances corresponding to the peaks showed that MEKO was detected in five peaks with retention times of 6.688 (peak 23), 6.744 (peak 24), 6.832 (peak 25), 6.986 (peak 27), and 7.037 (peak 28). Similarly, GC-MS analysis was performed on the remaining eight experimental groups. Due to space limitations, each set of tests was not analyzed in this paper and the experimental results are shown in the Supplementary Material (see Supplementary S1). Analyzing the nine sets of experiments together, the following conclusions were proposed. MEKO was detected in multiple peaks in experiments 1 to 8, but a small amount of MEKO was detected in only one peak in group 9. MEKOH was not detected in any of the nine experimental groups. The results of the nine sets of experiments are shown in Table 4.

**Table 4.** Analysis of the peaks appearing MEKO.

| Group | Peaks Belonging to MEKO | Retention Time | Area | Mass (MEKO) |
|---|---|---|---|---|
| 1 | 23 | 6.688 | 0.28 | 46<br>46<br>46 |
|  | 24 | 6.744 | 0.38 | 93<br>86<br>64 |
|  | 25 | 6.832 | 0.83 | 60<br>60 |
|  | 27 | 6.986 | 0.29 | 78<br>60<br>53 |
|  | 28 | 7.037 | 1.51 | 42<br>27 |
| 2 | 25 | 6.693 | 0.28 | 46<br>46<br>46 |
|  | 26 | 6.750 | 0.38 | 93<br>86<br>64 |
|  | 27 | 6.837 | 0.83 | 90<br>90<br>60 |
|  | 28 | 6.955 | 1.82 | 90<br>89<br>60 |
|  | 29 | 6.991 | 0.28 | 78<br>60<br>53 |
|  | 30 | 7.042 | 1.54 | 38<br>27 |
| 3 | 24 | 6.693 | 0.29 | 46<br>46<br>46 |
|  | 25 | 6.750 | 0.39 | 89<br>76 |
|  | 26 | 6.837 | 0.85 | 90<br>90<br>60 |
|  | 28 | 7.032 | 1.40 | 30<br>30 |
| 4 | 23 | 6.693 | 0.30 | 46<br>46<br>46 |
|  | 24 | 6.750 | 0.40 | 86<br>70 |
|  | 25 | 6.837 | 0.87 | 60<br>60<br>60 |
|  | 27 | 6.986 | 0.39 | 83<br>58 |

**Table 4.** *Cont.*

| Group | Peaks Belonging to MEKO | Retention Time | Area | Mass (MEKO) |
|---|---|---|---|---|
| 5 | 24 | 6.688 | 0.30 | 46<br>46<br>46 |
| | 25 | 6.745 | 0.41 | 86<br>64 |
| | 26 | 6.832 | 0.88 | 60<br>60<br>60 |
| | 28 | 6.996 | 0.56 | 83<br>64 |
| 6 | 24 | 6.693 | 0.39 | 46<br>46<br>46 |
| | 25 | 6.750 | 0.52 | 86<br>64 |
| | 26 | 6.837 | 1.12 | 90<br>78<br>60 |
| | 27 | 6.955 | 2.89 | 90<br>89<br>60 |
| 7 | 25 | 6.688 | 0.31 | 46<br>46<br>46 |
| | 26 | 6.745 | 0.43 | 93<br>90<br>76 |
| | 27 | 6.832 | 0.91 | 90<br>78<br>60 |
| | 28 | 6.924 | 1.34 | 52<br>52<br>46 |
| | 30 | 6.981 | 0.62 | 70<br>55<br>49 |
| 8 | 24 | 6.657 | 0.40 | 46<br>46<br>46 |
| | 25 | 6.714 | 0.54 | 93<br>86<br>64 |
| | 26 | 6.801 | 1.16 | 90<br>60<br>64 |
| | 27 | 6.904 | 2.68 | 50<br>50 |
| 9 | 17 | 6.411 | 0.49 | 42 |

The experimental results show that MEKOH is readily decomposed to MEKO, which is the reactant in the oximation reaction. This proves that the exothermic decomposition of MEKOH occurs as shown in Equation (2). Likewise, the conclusion provides theo-

retical support for optimizing the production process and improving the safety of the production process.

## 5. Discussion and Conclusions

In this experimental study, the thermal safety of MEKOH was comprehensively investigated from three perspectives: temperature changes during dissolution, the thermal stability of the substance, and analysis of pyrolysis products. Based on the experimental data and results, the following conclusions were obtained.

(1) The temperature profiles of MEKOH dissolved in different qualities of deionized water were measured and calculated with flask experiments, and a general warming pattern was determined. In the experimental environment of room temperature (28 °C), the final temperature of the MEKOH solution stabilized at about 33 °C, indicating an increase of approximately 5 °C throughout the entire process. During the process, MEKOH maintained a good thermal stability and was not found to violently decompose, which can be considered an acceptable temperature range. In other words, the dissolution of MEKOH at room temperature can be concluded to be relatively safe with no strong adverse effects on the safety of the production process.

(2) According to the comprehensive analysis results of TG analysis and DSC in the initial, intermediate, and complete states of MEKOH decomposition, MEKOH was found to have a good thermal safety below 50 °C. The substance underwent a violent exothermic decomposition from 51 to 57 °C, and no longer showed any significant change in weight after 145 °C. The enthalpy changes of the weight loss ranged from $-29.65$ J/g to $-45.86$ J/g. In actual production, the ambient temperature of MEKOH should be controlled below 50 °C to ensure the thermal safety of MEKOH and thus prevent flash explosion accidents.

(3) By analyzing the pyrolysis products, a large number of hydrocarbon compounds and other flammable substances were detected in the GC-MS experiments, but MEKOH was not detected. Once the temperature exceeded the temperature threshold for the thermal safety of MEKOH, the physicochemical properties of the pyrolysis products need to be promptly monitored and controlled while cooling. Moreover, a large amount of MEKO was found in the pyrolysis products of MEKOH, which is one of the reactants for preparing MOS. Therefore, MEKO from MEKOH decomposition can be recovered to be reused as a reactant, which can reduce a company's production costs and waste disposal hazards to some extent.

This experiment provides a monitoring basis and theoretical guidance for preventing the thermal runaway of MEKOH in the production process. However, there are a large number of other high-risk substances in the actual production system, and thermal behavior analyses and thermodynamic studies of other accident-prone substances and environments should be conducted in follow-up studies.

**Supplementary Materials:** The following supporting information can be downloaded at: https://www.mdpi.com/article/10.3390/su151914598/s1, Supplementary S1: GC and MS analysis of nine sets of experiments.

**Author Contributions:** Conceptualization and methodology, D.Z. and S.P.; validation, S.P., B.X. and L.W.; formal analysis, D.Z. and S.P.; investigation, L.W. and H.L.; data curation, D.Z.; writing—original draft preparation, D.Z.; writing—review and editing, S.P.; supervision, L.W.; project administration, B.X. and H.L. All authors have read and agreed to the published version of the manuscript.

**Funding:** This research was funded by the China Petroleum and Chemical Industry Federation (CPCIF) 2022 Responsible Care Special Research Priority Topics (grant number "2022CRCA001") and the Hubei Emergency Management Department 2021 Special Funds for Production Safety Project (grant number "Hubei Emergency development [2021] 18").

**Institutional Review Board Statement:** Not applicable.

**Informed Consent Statement:** Not applicable.

**Data Availability Statement:** All data used in the article and the relevant conclusions obtained therefrom are included in the article and Supplementary S1.

**Acknowledgments:** The authors sincerely acknowledge all the experimental participants for their efforts and contributions, as well as the Chemical Analysis and Testing Centre of Wuhan Institute of Technology for providing the experimental equipment.

**Conflicts of Interest:** The authors declare no conflict of interest.

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
