# Peer review of "Experimental Investigations of the Thermal Safety of Methyl Ethyl Ketone Oxime Hydrochloride Based on the Flask Method, Thermal Analysis, and GC-MS"

_sustainability, doi:10.3390/su151914598_

Round 1

Reviewer 1 Report

I have carefully read this paper entitled “Experimental investigations of the thermal safety of MEKOH based on flask method, thermal analysis, and GC-MS". As a result, it needs a major revision point that the authors should address before it is accepted for publication. There are a few grammatical mistakes. Please check the manuscript for grammar and English.

1-      Rewrite the novelty statement at the end of the introduction section.

2-      The abstract and conclusion should be rewritten and show the clearer result of this study.

Reviewer 2 Report

Comments: The introduction from lines 26-31 lacks cohesiveness and specificity. This whole section needs to be reframed.

The number of references for lines 31, 36 and in between lines needs to be relevant and needs to be increased. Also, please present a worldly scenario before focusing on China.

The take home message as well as the objective should be clearly mentioned in the concluding section of the introduction.

Fig.2: Please remove the horizontal lines from the diagram and use a colour on the greyscale.

Fig 3 is not clear. Increase the size of the legend.

Section 3: No references across the section. Please include relevant studies.

Results: The authors are advised to add the GC chromatogram for each of their studies. Also, they should clearly state as how they arrived at the retention time for each of the MEKOH samples.

also, please add a TEA for the number of studies that have been done so far.

The scientific english is not up to the mark. English must be checked by a native english speaker

Reviewer 3 Report

The manuscript titled Experimental investigations of the thermal safety of MEKOH based on flask method, thermal analysis, and GC-MS represents an important contribution in the field of occupational safety as a basis for sustainable development of the chemical industry. The research is extremely important because of the need to reduce accidents caused by thermal runaway of substances.

The abstract of the paper is clear and concise and written according to the instructions.

 > There are 47 literature references in the paper, which include all the research that has been conducted. However, the literature review should be in the introductory section, and the description of materials and experiments must clearly describe only the research conducted. In this way, the clarity of the experiment and the possibility of reproducibility are ensured. This must be corrected.

The results are clearly presented graphically using appropriate units and terminology and logically commented. The results show that methyl ethyl ketone oxime hydrochloride dissolved in deionized water increases the temperature by about 5 , which is an acceptable temperature range. When the temperature increases, methyl ethyl ketone oxime hydrochloride decomposes violently, releasing 29.65-45.86 J/g of heat. Through the comprehensive analysis of the thermal safety of methyl ethyl ketone oxime hydrochloride, the study provides theoretical evidence for the long-term work safety of enterprises MOS.

 > Also, the discussion of the results can be found in both the results and the conclusion. The discussion with the conclusions should be better argued and the conclusion should be shortened and more specific. This needs to be corrected.

 Kind regards, Reviewer

Round 2

Reviewer 2 Report

The authors have made significant improvements in the representation of the data. The manuscript can be accepted in the current format though there are major rooms for improvement.

Reviewer 3 Report

After the authors have corrected the work, I suggest accepting it in this form.

Rev. 3
